# Role of Ca^2+^ in Mediating Plant Responses to Extracellular ATP and ADP

**DOI:** 10.3390/ijms19113590

**Published:** 2018-11-14

**Authors:** Greg Clark, Stanley J. Roux

**Affiliations:** Department of Molecular Biosciences, University of Texas, Austin, TX 78712, USA; gblark@utexas.edu

**Keywords:** apyrase, calmodulin, nitric oxide, NADPH oxidase, reactive oxygen species

## Abstract

Among the most recently discovered chemical regulators of plant growth and development are extracellular nucleotides, especially extracellular ATP (eATP) and extracellular ADP (eADP). Plant cells release ATP into their extracellular matrix under a variety of different circumstances, and this eATP can then function as an agonist that binds to a specific receptor and induces signaling changes, the earliest of which is an increase in the concentration of cytosolic calcium ([Ca^2+^]_cyt_). This initial change is then amplified into downstream-signaling changes that include increased levels of reactive oxygen species and nitric oxide, which ultimately lead to major changes in the growth rate, defense responses, and leaf stomatal apertures of plants. This review presents and discusses the evidence that links receptor activation to increased [Ca^2+^]_cyt_ and, ultimately, to growth and diverse adaptive changes in plant development. It also discusses the evidence that increased [Ca^2+^]_cyt_ also enhances the activity of apyrase (nucleoside triphosphate diphosphohydrolase) enzymes that function in multiple subcellular locales to hydrolyze ATP and ADP, and thus limit or terminate the effects of these potent regulators.

## 1. Introduction

The growth and development of plants are controlled by a remarkably diverse array of endogenous chemical regulators beyond the hormones typically featured in textbooks, and new discoveries add to the list of these regulatory compounds every year. Among the most recently discovered of these compounds are extracellular nucleotides, especially extracellular ATP (eATP) and extracellular ADP (eADP) [1]. Although early studies showed that treatment of cells with these nucleotides could significantly alter the physiology of tissues and organisms, it wasn’t until the discovery of specific receptors for eATP and eADP that these purines became recognized as authentic endogenous regulators, first in animals, then in plants [2]. The purinergic receptor in plants [3], P2K1, is remarkably different from those identified in animals [4], yet its activation, like the activation of animal receptors, rapidly leads to a raised [Ca^2+^]_cyt_ [3,5,6,7], and downstream-increased levels of reactive oxygen species (ROS) and nitric oxide (NO). A focus of this review is to discuss the mechanisms that link receptor activation to increased [Ca^2+^]_cyt_, and the subsequent signaling steps that transduce the Δ [Ca^2+^]_cyt_ into major changes in plant growth and development. There is evidence to suggest that some eATP responses in plants may be initiated by a receptor other than P2K1, and this hypothesis is also discussed.

Just as the early signaling steps induced by eATP are similar in animals and plants, so too are the extracellular phosphatases that hydrolyze eATP and terminate its receptor activation [1]. Functionally dominant among these extracellular phosphatases are ectoapyrases (ecto-NTPDases). Among all the characterized extracellular phosphatases, ectoapyrases have the lowest Km for ATP hydrolysis, and their importance is highlighted by the fact that they are evolutionarily conserved in structure from algae through higher plants and mammals [1]. Their activity in the extracellular matrix (ECM) is closely associated with receptor activation since they remove the terminal phosphate only from NTPs and NDPs, which are the only nucleotides that can activate purinergic receptors. In plants, the overexpression or suppression of ectoapyrases has dramatic effects on gene expression and growth, and this review evaluates alternate pathways by which changes in apyrase expression could lead to transcription changes and hormone-transport changes that impact plant growth. Apyrase activity in plants is also regulated post-translationally by Δ [Ca^2+^]_cyt_, because two of the biochemically characterized plant ectoapyrases bind to and are stimulated by calmodulin (CaM) [8,9].

Although this review emphasizes Ca^2+^-dependent pathways by which extracellular nucleotides and apyrases regulate plant growth and development, there are also likely to be Ca^2+^-independent pathways to transduce eATP signals. Evidence for these alternative pathways is also presented and discussed. The literature on eATP signaling in plants is rapidly expanding, and concepts on how it modulates cell functions are in flux, so this review mainly features the data and insights contributed by more recent publications.

## 2. Rapid Changes in [Ca^2+^]_cyt_ Induced by eATP

Over a quarter of a century ago, it was established that, when eATP or eADP is functioning as a signaling agent in animal cells, one of the first changes it induces is an increase in [Ca^2+^]_cyt_ [10]. Thus, when scientists investigated whether plants had a similar response system, they tested whether the application of ATP could induce a rapid change in [Ca^2+^]_cyt_, and the positive results that were observed in cells of Arabidopsis root and shoot tissue [7,11] initiated an active search for purinergic receptors in plants. The search was biased at first to look for plant receptors that structurally resembled either one of the two well-characterized receptors in animals, i.e., the channel-linked P2X receptor or the G-protein-linked P2Y receptor [2].

The first plant receptor for eATP was discovered using a genetic approach and a screen based on whether mutants could show the Δ [Ca^2+^]_cyt_ response to applied eATP and eADP [3]. Surprisingly, it did not structurally resemble either of the animal receptors, but rather was a receptor kinase. This receptor, originally called Does Not Respond to Nucleotides1 (DORN1), and now designated P2K1, has a high binding affinity for both eATP and eADP, a transmembrane domain, and an extracellular nucleotide-binding domain [3]. This receptor is needed for eATP to transduce biotic stress stimuli into defense responses [3,12,13] and to regulate stomatal aperture [14]. More recently, Wang et al. [15] used patch-clamp electrophysiology to document that eATP rapidly activates both K^+^- and Ca^2+^-conductances in the plasma membrane of root epidermal cells, and this activation is dependent on P2K1.

The recent report by Chen et al. [14] was especially interesting in that it described a Ca^2+^-independent pathway by which P2K1 activation could induce an increase in ROS, which was by phosphorylating and activating the respiratory burst oxidase homolog protein D (RBOHD) subunit of the NADPH oxidase enzyme that synthesizes ROS. Because increased ROS can induce increases in [Ca^2+^]_cyt_ [16], this pathway suggested a mechanism by which eATP could induce an increased [Ca^2+^]_cyt_ that was very different from the way eATP induces this change by either of the receptors characterized in animals, because the P2X receptor induces increased [Ca^2+^]_cyt_ via opening a plasma membrane-localized ion channel, and the P2Y receptor does this by releasing Ca^2+^ from internal stores via G-protein and phosphoinositide mediation. However, two studies by Demidchik et al. [5,6] suggest that P2K1 may not be the only eATP/eADP receptor in plants, because they found that the kinetics of the induced increase in [Ca^2+^]_cyt_ differed depending on whether eATP or eADP was the extracellular nucleotide agonist. The rapidity of change induced by eADP (<3 s) more resembled the direct mechanism of cation channel opening when P2X is activated than an indirect mechanism that would require the intermediate steps of phosphorylation and synthesis of ROS.

More recently, Zhu et al. [17] found that the root-bending response away from unilaterally applied eATP, though dependent on Ca^2+^ uptake into the roots, occurs in *dorn1-1* and *dorn1-3* null mutants, indicating some other eATP receptor, as yet unidentified, is the initiator. Interestingly, this eATP response was dependent on the mediation of the G_α_ subunit of the heterotrimeric G-protein. Because this root-avoidance response was also blocked by EGTA and Gd^3+^, and because G_α_ activation is typically upstream of increased [Ca^2+^]_cyt_, this report implies a signaling pathway that begins with eATP activation of a non-P2K1 receptor, and proceeds through G_α_ activation and the opening of a plasma-membrane Ca^2+^ channel to a root-bending response. Like other root-bending responses, it was also dependent on asymmetric distribution of auxin and PIN2. This report is the first to document a P2K1-independent response to eATP, so discovering the receptor that mediates this response would be a novel and important contribution to the field. Although there are other reports of G-protein mediation of, or involvement in, the responses of plant cells to eATP [18,19,20], the fact that there are as yet no other reports of P2K1-independent responses to eATP, suggests that further studies would be needed to rigorously conclude that there are other eATP receptors in Arabidopsis besides P2K1.

An additional fact to consider regarding the question of the diversity of eATP receptors in plants is that, thus far, the only receptor found, P2K1, has only been characterized in Arabidopsis, and other plants may have receptors significantly different from this one. An ER-localized P2X-like receptor has been identified in algae [21]. That study justifies studies that are currently underway to determine whether the evolution of extracellular nucleotide receptors in plants may have followed more than one pathway, resulting in variety even more diverse than those found in animals.

## 3. Role of eATP-Induced Increase in [Ca^2+^]_cyt_ in Triggering Downstream Signaling Changes

After receptor activation by eATP, the earliest signaling changes detected in plant cells are changes in membrane potential and an increase in [Ca^2+^]_cyt_ [7,11,22]. Just as reported in animal cells, this increase in [Ca^2+^]_cyt_ can be observed as oscillations due to the transient opening of channels in the PM and/or on internal organelles, and, in Arabidopsis, these oscillations differ depending on whether the nucleotide agonist is ATP, GTP or CTP [19]. After the Ca^2+^ transport responses to eATP, the best-documented downstream steps are changes in the levels of ROS and NO. These changes occur in both animal and plant eATP signaling pathways [23] and can be causally linked to both the upstream changes in [Ca^2+^]_cyt_ [24] and the downstream physiological changes [17]. Moreover, they occur in response to many stimuli, and serve as regulatory agents in many aspects of plant growth and development as well as in abiotic and biotic stress responses [25,26,27]. The spatiotemporal aspects of these two second-messenger signals may allow for specificity. Their signaling activity is also affected by cross-talk from hormones and other environmental signals. Calcium and RBOHD-dependent ROS both produce systemic signals as waves that play a significant role in immune responses [28,29].

A recent study showed that eATP-activated P2K1 receptor directly phosphorylates NADPH oxidase RBOHD, causing an increase in ROS that leads to guard cell closure and increased resistance to *Psuedomonas syringae* [14]. This provides a direct link between the kinase activity of P2K1 and the production of the second messenger, ROS, apparently without the need for an intermediate step of increased [Ca^2+^]_cyt_. There are ROS-activated Ca^2+^ channels in plants [30], so the influx of Ca^2+^ across the plasma membrane induced by eATP could be a downstream effect of an earlier-induced increase in ROS. An early report implicated Arabidopsis annexin, AnnAt1, as playing a key role in this induced Ca^2+^ influx in root epidermal cells [31]. A subsequent study showed that AnnAt1 regulates H_2_O_2_-induced Ca^2+^ influx in roots [32]. The eATP-induced increases in ROS often lead to the activation of MAPKs [24]. The MAPK signaling pathway and RBOH-mediated Ca^2+^ signaling can also be regulated by Ca^2+^-dependent protein kinases during biotic stress responses [33,34].

Extracellular ATP plant responses are dose-dependent. Most of the ones studied thus far are biphasic, with lower concentrations of eATP or eADP inducing one response, and higher concentrations inducing the opposite response [35,36,37,38,39,40]. The differences in these dose-dependent responses are partially explained by the level of ROS and NO generated by different concentrations of applied eATP. For example, in Arabidopsis, the treatment of root hairs with 15–35 µM ATPγS promotes growth and induces low levels of ROS and NO, while treatment with ≥150 µM ATPγS inhibits growth and induces much higher levels of these two second messengers [37]. Making things even more complex, NO can rapidly react with ROS, specifically superoxide, to form peroxynitrite, a potent oxidant and nitrating species. Peroxynitrite can act as a signaling molecule, eliciting different cellular responses by meditating the post-translational nitration of tyrosine residues on target proteins [41,42]. Protein tyrosine nitration has been shown to be important in plant responses to both abiotic [43,44] and biotic stress [45], so, to the extent that eATP leads to the production of NO and peroxynitrite, this is another point of cross-talk in these stress-signaling pathways [46].

In tomato suspension-cultured cells, eATP-induced production of NO is dependent on the activation of phospholipase D and phospholipase C/diacylglycerol kinase, which catalyze the formation of phosphatidic acid [47], an important signaling intermediate in both biotic and abiotic stress responses [48,49]. In both tomato cells [47] and in hairy roots [50], the eATP-induced production of NO requires a prior activation of plasma-membrane Ca^2+^ channels. When NO production is induced by eATP in Arabidopsis, it appears to be generated by nitrate reductases, some of which are induced by increases in [Ca^2+^]_cyt_ [37]. NO can function as a signaling molecule by activating the enzyme guanylate cyclase, leading to the production of another second messenger, cGMP [51]. In both animal and plant cells, cGMP is sometimes converted to another signaling molecule, 8-nitro cGMP [52,53]. While cGMP mediates stomatal opening, 8-nitro cGMP was shown to induce stomatal closing [53], but is unclear if eATP-induced stomatal closing is mediated by 8-nitro cGMP.

Another way in which NO can act directly is by inducing nitrosylation of cysteine residues, a reversible post-translational modification that can regulate the activity of target proteins [54]. A key target protein modified post-translationally via an *S*-nitrosylation induced by NO is calmodulin. Regulation of calmodulin activity in this way would provide an avenue for cross-talk between NO and Ca^2+^ [51]. An NO–Ca^2+^ connection is also suggested by evidence that NO generation can lead to an increase in [Ca^2+^]_cyt_. The precise mechanism of this response and identity of Ca^2+^ channels involved remains unclear, but the regulation may occur by NO induction of cGMP [51].

Extracellular ATP acts like a hormone-like signal in plant cells, so it might be expected that eATP signaling pathways would lead to global changes in gene expression, similar to the action of other plant hormones. Several investigations into this question have provided evidence that it does indeed. An early study in Arabidopsis leaves revealed that treatment with eATP increased transcript levels of NADPH oxidase, respiratory burst homologue D, RBOHD, as well as lipoxygenase 2 and 1-aminocyclopropane-1-carboxylate synthase 6, genes involved in jasmonic acid and ethylene biosynthesis, respectively [24]. Exogenous ATP treatment of tobacco leaves induced differential expression of proteins involved in pathogen defense, photosynthesis, and oxidative stress [55]. Mutations that affect the expression level of apyrase enzymes that help control [eATP] also affect gene expression. This topic is covered in more detail in Section 5 below.

Consistent with the gene-expression changes observed in apyrase mutants, Choi et al. [3] documented that treatment with eATP upregulated the expression of genes that are also induced by wounding, especially those genes that respond early to wounding. Increases in [Ca^2+^]_cyt_ and calmodulin activation are known to have major effects on stress-induced changes in gene expression [56], so it is not surprising that at least some of the gene-expression changes induced by eATP are dependent on the early signaling changes induced by the binding of eATP to its P2K1 plasma-membrane receptor, which include an increase in [Ca^2+^]_cyt_. Choi et al. [3] showed that changes in the expression level of P2K1 in Arabidopsis differentially affected defense gene expression in response to wounding or eATP treatment. Treatment of cells with eATP also induced changes in the expression of genes involved in abiotic stress responses, as reported recently by Lang et al. [57]. They found that treatment of *Glycyrrhiza uralensis* roots with eATP increased the transcript levels of genes encoding salt overly sensitive 3, CBL-interacting protein kinase, respiratory burst oxidase homolog protein D, nitrate reductase, and the mitogen-activated protein kinases 3 and 6. The expression of the mitogen-activated protein kinases 3 and 6 was also enhanced by H_2_O_2_ in salt-stressed roots. These results highlighted the signaling interactions of eATP with Ca^2+^, H_2_O_2_, and NO in maintaining K^+^/Na^+^ homeostasis during salt stress in roots.

In this section, we reviewed publications that linked the early effects of receptor activation by eATP to downstream-signaling changes, revealing that the eATP-induced increase in [Ca^2+^]_cyt_ both precedes and is necessary for downstream signaling and physiological changes in plants. Figure 1 presents a model of a signaling pathway whereby early induction of the protein kinase activity of the P2K1 receptor by eATP results in the phosphorylation and activation of the RBOHD subunit of NADPH oxidase, which, in turn, sequentially leads to increased ROS, the opening of Ca^2+^ channels, and an increase in the [Ca^2+^]_cyt_. The model speculates there could be an alternative pathway whereby P2K1 receptor activation leads to the release of Ca^2+^ from internal stores, such as vacuoles and mitochondria. Based on the Zhu et al. [17] report, it also proposes the existence of another receptor that can activate a G_α_-dependent pathway to induce an increase in [Ca^2+^]_cyt_ that is required for the root-avoidance response to eATP.

## 4. Stress-Induced Growth Responses Regulated by eATP through Ca^2+^

Extracellular ATP acts as a hormone-like signal in a growing list of plant responses. As indicated previously, there are multiple ways for ATP to be released outside of plant cells, but the first and most obvious is via wounding [24]. Wound-released eATP acts as a signal in damage-associated molecular pattern (DAMP) responses that are induced by diverse biotic stresses [58]. These DAMP immune responses are regulated by cross-talk between hormones and eATP, and are mediated by increases in [Ca^2+^]_cyt_ and ROS levels [25]. Like eATP, the Ca^2+^ ionophore ionomycin induces both an increase in [Ca^2+^]_cyt_ and ROS levels, and, like eATP, has been linked to the induction of stress responses in plants [59].

Recent studies have characterized the interactions of eATP- and hormone-induced defense signaling and the points of cross-talk in these pathways. The role of the P2K1 receptor in mediating the eATP signal in Arabidopsis involves cross-talk with the defense hormone jasmonate (JA) [13,60]. Specifically, the expression level of P2K1 affects resistance to infection by *Pseudomonas syringae,* and it alters the defense gene-expression patterns mediated by JA and salicylic acid [60]. Activation of P2K1 by eATP promotes the activation of JA signaling, resulting in increased resistance to attack from the fungus *Botrytis cinera* [13]. This study showed that in addition to increasing the transcription of JA-related genes, eATP decreased the stability of jasmonate ZIM-domain 1 protein (JAZ1) in a Ca^2+^-, NO-, and ROS-dependent manner.

There is increasing evidence that, in addition to its role in biotic stress responses, eATP also plays a role in abiotic stress responses. In nature, these stresses often occur together and involve cross-talk from multiple hormones and signaling pathways [61,62]. Plant responses to abiotic stress are mediated by both NO and ROS [63]. An early study using cell cultures of the salt-tolerant woody plant *Populus euphratica* showed that salt treatment induced release of ATP. Furthermore, treatment with ATP resulted in a wide range of cellular responses needed for salt adaptation, and these responses were mediated by Ca^2+^ and H_2_O_2_ [64]. Several follow-up studies in *Populus euphratica* cells further characterized the eATP signaling pathway during salt stress. Zhao et al. [65] found that eATP stimulated Na^+^ extrusion and reduced K^+^ ion loss during salt stress in a Ca^2+^- and H_2_O_2_-dependent manner. Another study found that salt-induced vacuolar Ca^2+^ release was mediated indirectly by eATP, NO and ROS [66]. Interestingly, the roles of eATP, Ca^2+^, H_2_O_2_, and NO in mediating root ion fluxes were different in secretor and nonsecretor mangrove seedlings during salt stress [67]. Gene-expression changes related to K^+^/Na^+^ homeostasis are also affected by eATP during salt-stress responses [57].

Cold treatment is another form of abiotic stress that can induce both the release of ATP to the ECM and an influx of Ca^2+^ from external stores as two of the earliest cellular signals [68]. In this response to cold stress, the increase in [eATP] appears to help in the repair of cold-damaged membranes, as shown by Deng et al. [40]. They found that treatment with low levels of eATP promoted the vesicular trafficking needed for membrane repair, while treatment with high levels of eATP inhibited trafficking [40]. This study also found that Arabidopsis seedlings ectopically expressing an apyrase from *Populus euphratica* had enhanced cold tolerance, as determined by improved plasma-membrane integrity and root growth compared to the wild type in response to cold stress.

Extracellular ATP can also play a role in mediating the effects of heavy-metal poisoning that would occur, for example, by treatment with the heavy metal cadmium. In pea plants, response to cadmium stress involves cross-talk between Ca^2+^, ROS, and NO, all of which are also downstream signals in eATP signaling pathways [69]. Cadmium stress induces the release of ATP in Arabidopsis leaves [70], which then show the characteristic negative effects of increased levels of lipid peroxidation as well as higher lipoxygenase and antioxidant activities. A competitive inhibitor of eATP partially dampens these effects. Similarly, the P2K1 loss-of-function mutant showed lower levels of lipid peroxidation and antioxidant activities compared to the wild type, indicating that the metabolic changes induced by cadmium stress are mediated in part by the ATP released by stressed cells.

When plants respond to abiotic and biotic stresses, their growth is typically inhibited. This may be part of the explanation for the decreased growth observed in plant cells and tissue when treated with high levels of exogenous ATP [23]. As previously indicated, plant-growth responses to applied ATP or ATPγS are dose-dependent, with low levels promoting and high levels inhibiting growth in a wide variety of tested cells and tissue, including hypocotyls, root hairs, pollen tubes, and cotton fibers [23]. When testing the effects of nucleotides on plant responses, scientists often use poorly hydrolyzable analogs of ATP and ADP, ATPγS and ADPβS, to avoid the potentially confounding effects of released phosphate when ATP or ADP are hydrolyzed. Because of the slower turnover of ATPγS and ADPβS in the ECM, treatment with approximately 10-fold less of these agents elicits the same growth response as treatment with higher levels of ATP and ADP. Inhibition of growth is also induced by raising the [eATP] via suppression of ectoapyrase activity genetically, chemically, or immunologically [37,38,71,72]. Correspondingly, promotion of growth is also observed by lowering the [eATP] via overexpressing or ectopically expressing ectoapyrases in plants [71].

A potential explanation of how eATP can regulate plant growth may be derived from the observation that eATP affects transport of the growth hormone auxin [73,74]. In Arabidopsis, high levels of applied ATP and suppression of ectoapyrase expression inhibit basipetal auxin transport, while overexpression of ectoapyrase promotes basipetal auxin transport [74]. In most examples of eATP regulation of growth, there is also evidence that it is mediated by the downstream changes in Ca^2+^, NO, and ROS. For example, eATP effects on hypocotyl and root-hair growth in etiolated Arabidopsis seedlings was dependent on NADPH oxidase activity and NO production [75,76]. The eATP-induced increase in NO and ROS production have been plausibly linked to prior eATP-induced increases in [Ca^2+^]_cyt_ because of the Ca^2+^ dependence of enzymes that generate NO and ROS [1,24]. Furthermore, mutants null for the P2K1 receptor fail to show both increased [Ca^2+^]_cyt_ and downstream signaling and gene-expression changes in response to eATP [3]. As a further test of the link between the eATP-induced increase in [Ca^2+^]_cyt_ and downstream cellular responses, Song et al. [24] showed that Ca^2+^-channel blocker (LaCl_3_), Ca^2+^ chelator (BAPTA), and calmodulin antagonist (W7) all blocked the eATP-induced increase in ROS.

In addition to regulating primary root growth, the eATP signaling pathway affects other aspects of root growth and development in Arabidopsis seedlings. For example, treatment with eATP or suppressing the expression of AtAPY1 affects the root skewing growth response in Arabidopsis [77]. The eATP-induced Ca^2+^ influx and root-skewing response are reduced in the loss-of-function mutants for the H^+^-ATPase (AHA2). This result indicates that the plasma-membrane proton motive force plays an important role in this eATP response [78]. Consistent with this result, this study also showed that eATP-induced root skewing in wild-type seedlings was pH-dependent. In addition to their use in the studies noted above on how extracellular nucleotides affect plant-cell responses to abiotic stress, plant-cell cultures have also been used to reveal a role for eATP in cell viability and in programmed cell death (PCD). Experiments in Arabidopsis cell cultures showed that eATP was needed for maintaining plant-cell viability [79]. In contrast, in *Populus euphratica* cell cultures, applied eATP induced PCD in a dose- and time-dependent manner [80], and in tobacco cell cultures, eATP appeared to affect salicylic acid-induced PCD [81,82]. These studies employed prolonged exposures of cells to high levels of eATP, so, given that there are enzymes in the ECM that can rapidly convert eATP to ADP and AMP, and other enzymes that can convert AMP to adenosine and cAMP, the role of eATP in the effects reported may be indirect, serving as a substrate for the production of other metabolites that were the more direct mediators of these effects.

Most of the studies on eATP-induced effects on growth and stress responses in intact plants previously described in this section were done on aerial tissues or on primary roots of seedlings. However, the regulatory effects of eATP may also occur in more mature stages of plant development, such as in fully expanded leaves. In these types of tissue, there are several reports of eATP effects on photosynthesis and on stomatal function. Several recent studies have indicated that exogenously applied ATP stimulates photosynthesis. For example, in kidney-bean leaves, treatment with 1 mM ATP enhances photosystem II (PSII) photochemistry, and this stimulation is dependent on Ca^2+^ and H_2_O_2_ [83]. In Arabidopsis leaves, where hypertonic salt treatments release ATP, the negative effects of these treatments on PSII and intracellular ATP production are more severe in a P2K1 loss-of-function mutant [84]. In another example of eATP effects on PSII, eATP alleviates the negative effects on PSII in *Phaseolus vulgaris* leaves caused by infection by a pathogenic bacterium [85].

Like growth, changes in stomatal aperture requires changes in the plasma-membrane surface area, which are mediated by exocytosis and endocytosis [86]. Because exocytotic secretory vesicles contain ATP that gets released to become eATP, regulation of stomatal aperture was considered another possible eATP-regulated response. An early study in Arabidopsis found that low [eATP] induced stomatal opening, while high [eATP] induced stomatal closing and that ectoapyrases, ROS, and NO played a role in eATP-mediated changes in stomatal aperture [39]. This study also found that ABA- and light-induced changes in stomatal aperture were accompanied by corresponding changes in the [eATP]. Another Arabidopsis study documented that applied ATP induced Ca^2+^ influx and H^+^ efflux during stomatal opening. This study also reported a key role for heterotrimeric G protein in eATP-induced stomatal opening [20]. There is also evidence that the heterotrimeric G protein also participates in eATP-induced stomatal closing [87]. Consistent with this finding, Wang et al. [88] found that the eATP-induced guard cell opening in *Vicia faba* was dependent on ROS production, which appeared to activate a hyperpolarization-activated Ca^2+^ channel on the plasma membrane.

Stomatal closing is a central response in innate immunity [89], so given the role of eATP in defense responses, it is not surprising that the high levels of eATP released in DAMP responses induce stomatal closure. The P2K1 receptor for defense response was also determined to be the receptor for eATP-induced guard cell closing [14]. eATP was also found to play a role in hydrogen sulfide-induced stomatal closure, and eATP induced closure by promoting K^+^ channel activity and NADPH oxidase-dependent ROS production [90]. Interestingly, this study also found that hydrogen sulfide treatment induced the release of eATP via ABC transporters.

## 5. Role of Ca^2+^/CaM-Activated Apyrase in Mediating Growth and Development Changes in Plants

Both in animals and plants, the main extracellular phosphatase that limits the concentration of extracellular nucleotides is ectoapyrase (ecto-NTPDase) [91]. As reviewed by Clark et al. [1], ectoapyrases have been characterized in diverse plants, including multiple legumes, Arabidopsis, poplar, and potatoes. All apyrases require divalent cations for activation, for which the basal levels of Ca^2+^ and/or Mg^2+^ in cells would typically suffice [87]. However, the activity of two well-characterized apyrases, psNTP9 and AtAPY1, are CaM-activated, and their activity would be expected to increase more than threefold whenever an external stimulus increased the [Ca^2+^]_cyt_ above the 100 nM level needed to activate CaM [8,9]. Thus, at least these two apyrases, and probably others not yet characterized, must be listed among the key enzymes whose activity is increased by Ca^2+^-acitvated CaM.

To examine the function of ectoapyrases in plants, transformation studies were carried out. The first ectoapyrase used in these transformation studies was psNTP9, which Thomas et al. [92] showed could promote phosphate uptake and leaf growth when it was expressed in Arabidopsis seedlings. More recent studies, which examined the effects of ectopically expressing psNTP9 on mature plants, showed that it improved phosphate uptake, leaf growth, root-system architecture, and seed yield in both Arabidopsis and soybean plants [93]. Genetic manipulation of ectoapyrase expression in Arabidopsis has been carried out in numerous other studies. As noted in Section 4, the suppression of the expression of two closely related apyrases in Arabidopsis, AtAPY1 and AtAPY2, which have been proposed to function as ectoapyrases [1], blocked pollen germination [94] and strongly suppressed growth [72,95]. Conversely, overexpressing either of these two native apyrases in Arabidopsis promoted growth [71] and enhanced the transport of the growth hormone auxin [74]. Another study of ectopically expressing an apyrase in Arabidopsis was done using a poplar apyrase (PeAPY2), and it showed that this apyrase could enhance cold tolerance in Arabidopsis seedlings [40].

Besides the report of Veerappa et al. [93] noted above, several other transformation studies have been used to explore the function of apyrases in crop plants. The overexpression of an endogenous apyrase in *Lotus japonicus* [96] promoted nodulation, and the suppression of the GS52 apyrase in soybean inhibited nodulation [97]. Taken together, these two reports strongly supported the conclusion that apyrase expression helped to regulate nodulation in legumes. The RNAi-induced suppression of apyrase in potatoes decreased growth rate and increased the number of tubers per plant [98].

Ectoapyrases also play a role in defense responses. For example, in peas, a calmodulin-regulated ectoapyrase forms a protein complex with copper amine oxidase, an enzyme that is involved in extracellular H_2_O_2_ production during the response to a fungal attack by *Mycosphaerella pinodes* [99]. Interestingly, this ectopapyrase appears to be the target of elicitor and suppressor molecules secreted by the fungus. Extracellular ATP and the ectoapyrases that hydrolyze eATP also appear to be a battleground between plants and insects during defense responses [100,101,102].

The discovery of apyrase-like activity in purified pea nuclei that were inhibited by the CaM antagonist compound 48/80 led Chen et al. [103] to test whether a CaM-regulated apyrase could be purified from pea nuclei, and their positive results indicated that Ca^2+^-activated CaM could stimulate the ATPase activity of the purified apyrase threefold [8]. Hsieh et al. then identified the CaM binding region on a recombinant pea apyrase expressed in *Escherichia. coli* (residues 293–308) and showed that this binding inhibited the in vitro phosphorylation of the enzyme by a recombinant human protein kinase CKII [104]. Steinebrunner et al. [9] then characterized the two closely related apyrases in Arabidopsis, and found that only one of them (AtAPY1) could bind calmodulin and be activated by it.

The psNTP9 apyrase from pea nuclei characterized by Chen et al. [8] and Hsieh et al. [104] was independently characterized under a different name, PsAPY1, by Shibata et al. [105,106], who found it to also be associated with the cell wall and the cytoskeleton, as well as with the nucleus. The wall localization had earlier been confirmed by Thomas et al. [92], and the immunofluorescence assays of Tong et al. [107] revealed significant signals in both nuclei and wall regions of etiolated pea seedlings. Evidently, psNTP9 can either be an ectoapyrase or in different endodomains of the cell depending on environmental cues and stage of development.

The increased [Ca^2+^]_cyt_ induced by eATP would differently impact CaM-regulated apyrases, like psNTP9 and AtAPY1, depending on their subcellular locale. There is CaM in the wall domain of plants [108,109], but an increase in [Ca^2+^]_cyt_ would not be expected to directly alter the activity of calmodulins in the ECM, where [Ca^2+^] would typically be above the threshold needed to activate CaM [110]. An increase in [Ca^2+^]_cyt_, however, would likely increase the [Ca^2+^] of the nucleus, since the [Ca^2+^] between these two cell compartments appears to rapidly equilibrate [111]. The activation of nuclear CaM would result in an increase in APY activity there, which would alter nuclear NTP concentrations, and this change would be expected to have significant impacts on multiple NTP-dependent nuclear activities, including transcription, protein phosphorylation, chromatin remodeling [112], and transcript splicing [113].

Massalski et al. [114] verified that AtAPY1 was activated by CaM, but they also found that the enzyme was primarily an ADPase and could not use ATP as a substrate. They used different modifications of AtAPY1 than those used in an earlier report [9], which had found that AtAPY1 favored ATP as a substrate. These differences may have altered the substrate specificity of the enzyme. One of the versions of AtAPY1 used in the more recent report was isolated from Arabidopsis and tagged with GFP, while the other was purified from Human Embryonic Kidney cells (HEK293) and included only residues 67–470 after the N-terminal transmembrane domain [114]. In contrast, the AtAPY1 assayed in the earlier report [9] was His-tagged and expressed in *E. coli*. Certainly, GFP tags can alter the activity of an enzyme [115], as can any other modification of its primary structure. To help resolve the substrate specificity of AtAPY1, Weeraratne purified the native enzyme without any affinity tags to near homogeneity from the nuclei of etiolated Arabidopsis seedlings and she found, in agreement with the earlier report [9], that AtAPY1 favored ATP as a substrate (Weeraratne, PhD dissertation, 2018).

The primary structures of AtAPY1 and AtAPY2 are 87% identical, and these two apyrases partially complement each other’s function [1]. Although the straight growth of primary roots is the same in mutants knocked out or overexpressing either of the genes encoding these enzymes, only mutations that alter the expression of *APY1* significantly change the root-skewing responses [77]. These results indicate that APY1 plays a more important role in regulating root skewing than APY2. Almost certainly there are other functional differences between these two apyrases, and it will be useful to determine whether the fact that Ca^2+^-activated CaM affects the activity only of AtAPY1 accounts for any of the functional differences observed.

## 6. Conclusions and Perspectives

Changes in [Ca^2+^]_cyt_ are characteristically early steps in the signaling pathways induced by most plant growth regulators, so it is not surprising that it is also one of the first signaling changes induced by eATP. This review describes evidence that eATP-induced changes in [Ca^2+^]_cyt_ both precede and are necessary for the downstream signaling changes that promote the physiological responses to eATP. Turning off signaling pathways is as important as turning them on and, in peas and Arabidopsis, the same Δ [Ca^2+^]_cyt_ that turns on the downstream increases in ROS, NO, and MAP kinase activity also, through CaM, activates an apyrase that removes the eATP agonist, thus inactivating the receptor. Although the connection between receptor activation and a rapid increase in [Ca^2+^]_cyt_ is well understood in mammalian cells, we note in this review that more research is needed to resolve this connection in plants.

Another significant limitation in the current state of knowledge about eATP signaling is that too much of this knowledge is based on research on only one plant, *Arabidopsis thaliana*, a member of the Brassicaceae family, which includes less than 1% of the flowering plant species. Other families of plants may have other kinds of receptors, and certainly some plants may not have any CaM-regulated apyrases. The discovery of the P2K1 receptor for eATP has spurred increased interest in purinergic signaling, and this new wave of interest will surely lead to broadening the scope of research to include other model plants, both vascular and nonvascular.

## Figures and Tables

**Figure 1 ijms-19-03590-f001:**
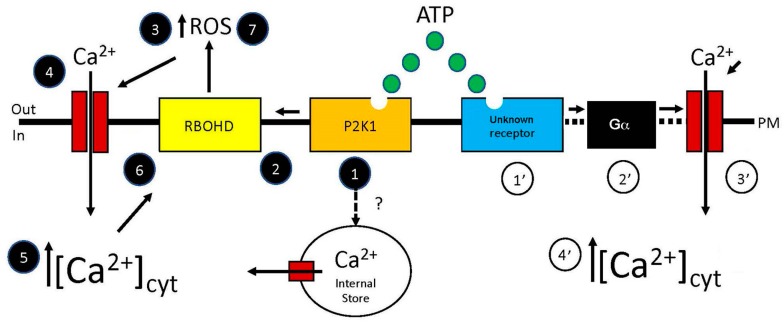
Model of two hypothetical signaling pathways whereby eATP could induce an increase in [Ca^2+^]_cyt_. In this model, two potential receptors could initiate the signaling pathway. The one starting with P2K1 is much better documented and supported by published data, whereas the one starting with a postulated (and as yet unknown) receptor is supported only by indirect evidence. The sequence of signaling steps initiated by P2K1 would include: 1, the binding of ATP to the plasma membrane-localized P2K1 receptor, thus activating its protein kinase activity; 2, the receptor-induced phosphorylation and activation of the RBOHD subunit of NADPH oxidase; 3, increased accumulation of reactive oxygen species (ROS) in the extracellular matrix (ECM); 4, ROS-induced opening of calcium channels; 5, increase in the [Ca^2+^]_cyt_. This increase could then further activate RBOHD to synthesize more ROS (Steps 6 and 7). The sequence of signaling steps initiated by the postulated unknown receptor would include: 1′, the binding of ATP to the receptor; 2′, activation of G_α_; 3′, the direct or indirect induction of Ca^2+^ channel activity by G_α_; and 4′, increase in the [Ca^2+^]_cyt_. There is also evidence for an as-yet unidentified pathway whereby receptor activation leads to the release of Ca^2+^ from internal stores, such as vacuoles and mitochondria. Dotted lines indicate steps mediated by as-yet unknown mechanisms.

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
