# Peer review of "Role of Ca2+ in Mediating Plant Responses to Extracellular ATP and ADP"

_ijms, 2018, doi:10.3390/ijms19113590_

Reviewer 1 Report

This review presents a fairly wide overview of recent literature on eATP signaling in plants.

However, based on the title of this review - Role of Ca2+ in mediating plant responses to  extracellular ATP and ADP – I expected this review to focus on:

1 Characteristics of ATP-induced cytosolic Ca changes (spatio-temporal features, upstream steps and transport mechanisms involved)

2 Evidence that such cytosolic Ca change has a role in plant response to eATP and eADP. To say it with Jaffe: i) does it occur prior to the physiological response? ii) is the physiological response abolished if cytosolic Ca change is suppressed or altered? iii) can the response be elicited by an artificially induced cytosolic Ca change?

3 Which are the consequences of the  cytosolic Ca change?

Rather, I found a summary of recent work on eATP signaling in which Ca is not even mentioned for pages and the information on the above listed points is scattered throughout the text. Moreover, the attempt to recall the Ca focus make the summary rather confused.

Reviewer 2 Report

The manuscript by Clark and Roux updated the knowledge of extracellular ATP and ADP roles in plant growth and regulation. They summarised how eATP and eADP excite the downstream signalling actions, such as cytosolic calcium signalling, NO and ROS signalling, and how these downstream signals are further turnover into plant response. The whole review is well described. So I would recommend it accepted by IJMS and only suggest a minor changes

Line 235, maybe briefly a few words explain what ATPγS is and how commonly it is used to replace ATP treatments.

Authors' Response to Reviewers’ comments:

Response to Reviewer 1

All of these comments and recommendations of Reviewer 1 were valid and valuable, and in the four sentences below we describe how the revised text accommodates these comments. As a result, the review is much improved.  
1. We added text on pages 11, 12, 16, and 20 to focus more on the role of calcium in eATP signaling, so this focus now is more evident throughout the text.
2.We added text on pp. 5, 6, 10, 11, 13, and 14 to document (with references), in accord with Jaffe criteria, that (i) cytosolic Ca2+ changes occur prior to the physiological response; ii) that the physiological response is abolished if cytosolic Ca change is suppressed or altered; and, iii) the response can be elicited by an artificially induced cytosolic Ca change.
3. We also added text on pages 4, 5, 6, 10 and 15 describing in more detail the spatio-temporal features, upstream steps and transport mechanisms involved in ATP-induced cytosolic Ca2+ changes.  
4. As a result of the changes noted above, the consequences of the cytosolic Ca change for downstream signaling changes (ROS, NO), and physiological changes (e.g., stress responses) are more evident in the review.

Response to Reviewer 2

We appreciate this recommendation, and we have added text on pp. 13-14 explaining that ATPγS is a poorly hydrolyzable ATP analog that is often used in place of ATP treatments. 

Round  2

Reviewer 2 Report

This revised manusript updated a few more recent studies, this add-up makes the research in this field thoroughly discussed. I would recommend it accepted by IJMS.

Other Reviewers'  Report

I have reviewed the manuscript and checked the reponses of the authors to the two reviewers. I consider that the authors have addressed all concerns of the two reviewers and the manuscript is ready to be published.